# Creating a Multimodal Translation Tool and Testing Machine Translation Integration Using Touch and Voice

**Carlos S. C. Teixeira [1,*] , Joss Moorkens [2] , Daniel Turner [3], Joris Vreeke [3] and Andy Way [3]**

1   IOTA Localisation Services/Trinity Centre for Literary and Cultural Translation, Trinity College Dublin, D02 CH22 Dublin, Ireland
2   ADAPT Centre/School of Applied Language and Intercultural Studies, Dublin City University, D09 Y074 Dublin, Ireland; joss.moorkens@dcu.ie
3   ADAPT Centre/School of Computing, Dublin City University, D09 Y074 Dublin, Ireland; dturner@tcd.ie (D.T.); joris.vreeke@adaptcentre.ie (J.V.); andy.way@adaptcentre.ie (A.W.)
*   Correspondence: carlostx@linguanativa.com.br

**Abstract:** Commercial software tools for translation have, until now, been based on the traditional input modes of keyboard and mouse, latterly with a small amount of speech recognition input becoming popular. In order to test whether a greater variety of input modes might aid translation from scratch, translation using translation memories, or machine translation postediting, we developed a web-based translation editing interface that permits multimodal input via touch-enabled screens and speech recognition in addition to keyboard and mouse. The tool also conforms to web accessibility standards. This article describes the tool and its development process over several iterations. Between these iterations we carried out two usability studies, also reported here. Findings were promising, albeit somewhat inconclusive. Participants liked the tool and the speech recognition functionality. Reports of the touchscreen were mixed, and we consider that it may require further research to incorporate touch into a translation interface in a usable way.

**Keywords:** computer-aided translation; usability; agile development; multimodal input; translation technology

## 1. Introduction

Computer-Assisted Translation (CAT) tools, for the most part, are based on the traditional input modes of keyboard and mouse. We began this project with the assumption that multimodal input devices that include touch-enabled screens and speech recognition can provide an ideal interface for correcting common machine translation (MT) errors, such as word order or capitalization, as well as for repairing fuzzy matches from translation memory (TM). Building on our experience in soliciting user requirements for a postediting interface [1] and creating a prototype mobile postediting interface for smartphones [2], we aimed to create a dedicated web-based editing environment for translation and postediting with multiple input modes. On devices equipped with touchscreens, such as tablets and certain laptops, the tool allows translators to use touch commands in addition to the typical keyboard and mouse commands. On all devices, the tool also accepts voice input using system or standalone automatic speech recognition (ASR) engines. We considered that translation dictation would become more popular, and that the integration of touch would be helpful, particularly for repairing word order errors or dragging proper nouns (that may present a problem for ASR systems) into the target text.

Another important aspect of the tool is the incorporation of web accessibility principles from the outset of development, with the aim of opening translation editing to professionals with special needs.

In particular, the tool has been designed to cater for blind translators, whose difficulty in working with existing tools has been highlighted by Rodriguez Vazquez and Mileto [3].

This article reports on usability tests with sighted (nonblind) professional translators using two versions of the tool for postediting. The results include productivity measurements as well as data collected using satisfaction reports and suggest that voice input may increase user satisfaction. While touch interaction was not initially well-received by translators, after some interface improvements it was considered to be promising. The findings have also provided relevant feedback for subsequent stages of iterative development.

## 2. Related Work

In previous work by our team, we investigated user needs for a postediting interface, and found widespread dissatisfaction with current CAT tool usability [1]. In an effort to address this, we developed a test interface in order to see what benefits some of the suggested functionality might show [4]. We also developed a touch- and voice-enabled postediting interface for smartphones, and while the feedback for this interface was very positive, the small screen real estate presented a limitation [5]. The development reported here was a natural continuation of the smartphone project.

Commercially available CAT tools are now starting to offer integration with ASR systems (e.g., MemoQ combined with Apple's speech recognition service and Matecat combined with Google Voice). To the best of our knowledge, however, there are no independent research studies on the efficiency and quality of those systems, or on how translators perceive the experience of using them. With other CAT tools, it is possible to use commercial ASR systems for dictation such as Dragon Naturally Speaking, although the number of languages available is limited and the integration between the systems is not as effective as users might expect [6].

Research on the use of ASR in noncommercial CAT tools includes Dragsted [7] and Zapata, Castilho, & Moorkens [8]. The former reports on the use of dictation as a form of sight translation, to replace typing when translating from scratch, and the latter proposes using a segment of MT output as a translation suggestion. In the present study, we used ASR as a means of postediting MT proposals propagated in the target text window (and in the future intend to incorporate ASR for repairing suggestions from TMs). These activities require text change operations that are different from the operations performed while dictating a translation from scratch.

One study that investigates the use of ASR for postediting is by Garcia Martinez et al. [9] within the framework of the Casmacat/Seecat project. For the ASR component, the authors combined information from the source text through MT and semantic models to improve the results of speech recognition, as previously suggested by Brown et al. [10], Khadivi & Ney [11], and Pelemans et al. [12].

Different interaction modes have been explored in the interrelated projects Caitra [13], Casmacat [14], and Matecat [15], with a particular focus on interactive translation prediction and smarter interfaces. In Scate [16] and Intellingo [17], the main area of investigation has been the presentation of "intelligible" metadata for the translation suggestions coming from MT, termbases, and TMs, as well as experiments for identifying the best position of elements on the screen.

In terms of other modes of interaction with the translation interface, the closest example to the current study is the vision paper by Alabau & Casacuberta [18] on the possibilities of using an e-pen for postediting. However, that study is of a conceptual nature and did not test the assumptions of how an actual system would perform.

In the study presented in this article, we used a publicly available ASR engine (Google Voice), which connected to our tool via an application programming interface (API). For touch interaction we used only the touchscreen operated with the participants' fingers, with no additional equipment such as an e-pen. Our approach differs from some of the studies mentioned above in that we moved from the conceptual framework to actually testing all of those features in two rounds of tests with 18 participants in total.

### 3. Tool Development

The main objective of the research & development project reported on in this article was to develop a fully-fledged desktop-based CAT tool that was at the same time multimodal and web- accessible, as explained in the previous section. By multimodal we mean a tool that accepts multiple input modes, i.e., that not only allows translators to use the traditional keyboard and mouse combination, but that also supports touch and voice interactions. By web-accessible we mean a tool that incorporates web accessibility features/principles, for compatibility with assistive technologies in compliance with W3C web accessibility standards and the principles of universal design [19], as will be explained in more detail later. We also wanted the tool to be compatible with the XLIFF standard for future interoperability with other tools, and to be instrumented (i.e., to log user activities), to facilitate the automatic collection of user data.

From a development point of view, the project began in March 2017 and has progressed using an Agile/Scrum process of short sprints. An initial decision was made to create a browser-based tool, as this would work on any operating system, would not require download and installation, and follows a trend towards software-as-a-service. The planning phase involved sifting through the available frameworks and providing a breakdown of each. After analysis, the proposed solution was to use React, Facebook's component-based library for creating user interfaces. Using a modern library like React exposes the tool to a large repository of compatible, well-developed technologies which greatly increase the ease and speed of development. An example of this is Draft JS, the rich text editor implemented by the tool. React provides the core functionality for building the web-based interface, allowing the tool to be used across platforms. Moreover, React Native provides an implementation of React for building native applications, which opens the potential for the tool to be ported over to iOS and Android with minimal development costs.

Although the tool can run on any browser, it has been tested most extensively on Google Chrome as we are using the Google Voice API as the connected ASR system, and this is only available on Google Chrome. The tool may be connected to other ASR systems as required.

### 4. The Initial Prototype

The tool offers two different views for producing translations. In the Edit view, translators can either use the keyboard and mouse or they can click a microphone icon and dictate their translations. In the Tile view, each word is displayed as a tile or block, which the translator can then move, edit, etc. using the touchscreen with their fingers.

Figure 1 shows the Edit view in our initial prototype. The source text appears in the center with a suggestion from MT below, then a bar with formatting options followed by a box where the target text may be entered. One may notice that we chose to offer a 'minimalist' interface based on a current trend in other web-based CAT tools as well as on (as yet inconclusive) studies that indicate that standard interfaces are not optimally designed from the perspective of cognitive ergonomics [20]. This may be compared with the second iteration of the Edit view in Figure 13.

Figure 2 shows additional features present in the tool: Comments, Lexicon (making use of the BabelNet encyclopedic dictionary [21]), and Global Find and Replace. Although these features have been made available from the early development stages, they have not yet been included in a test scenario.

Figure 3 shows the dictation box that opens when clicking the microphone button. When the box is displayed, the translator can click the second microphone icon in the box to activate speech recognition. Once the dictation has been completed, the result of the recognition appears in the corresponding area. If translators think the ASR output is useful, they can accept it by clicking the green tick; if they are not satisfied with the output, they can click the X button to clear the output and dictate again. Translators can also edit the ASR output in the dictation box before accepting it.

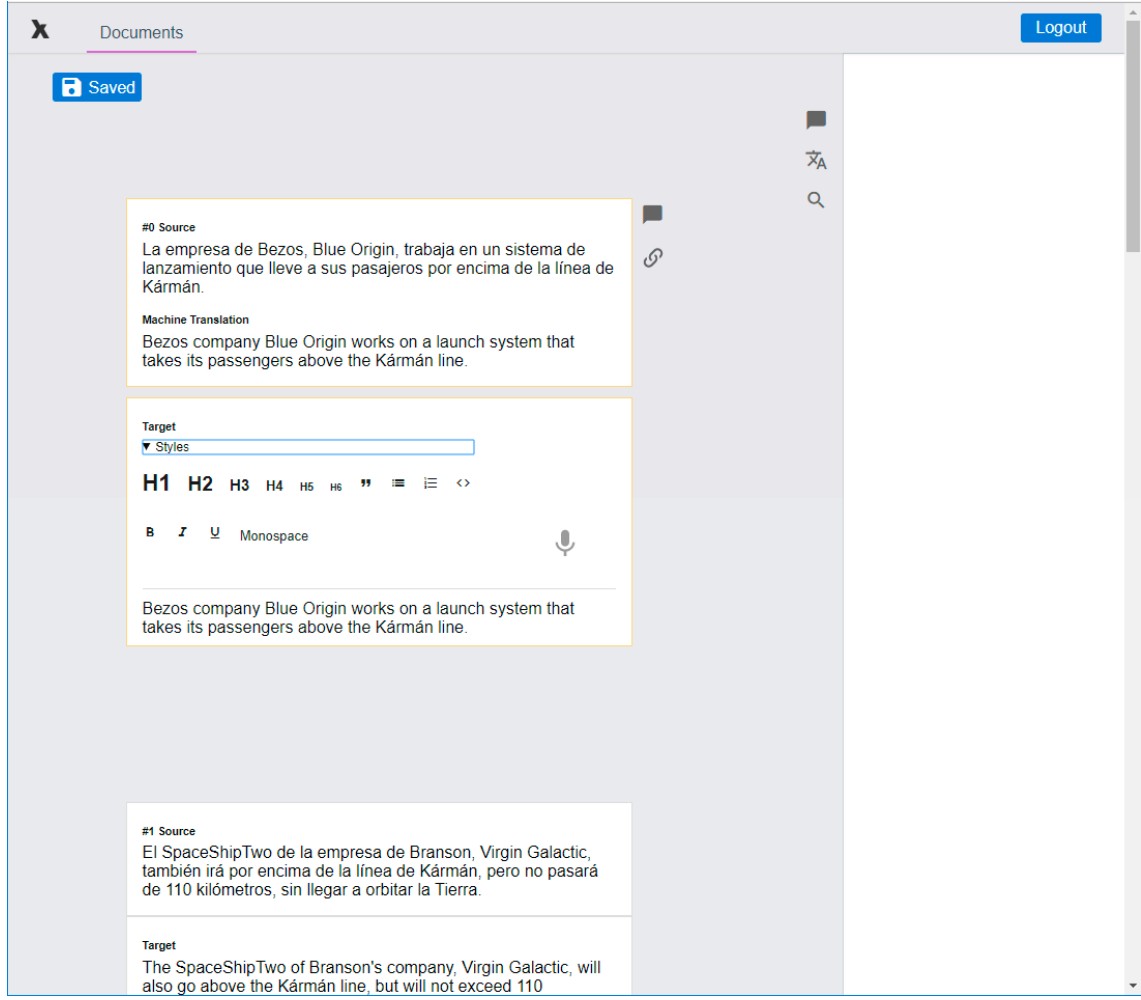

**Figure 1.** Interface of our initial prototype showing the basic Edit view.

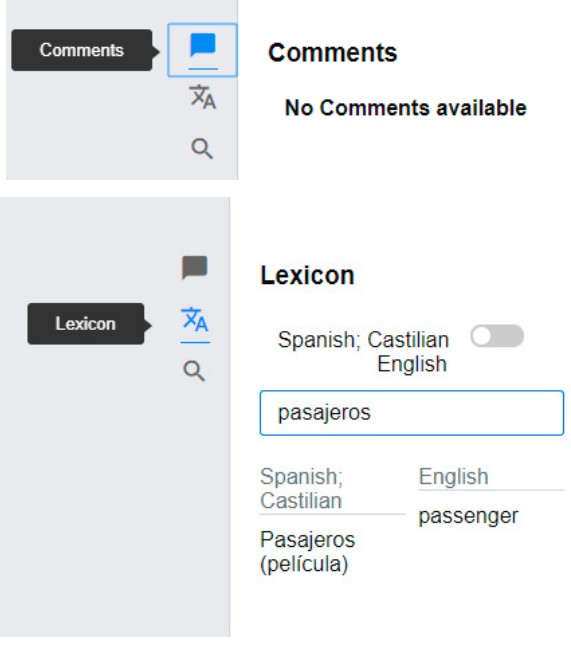

**Figure 2.** *Cont.*

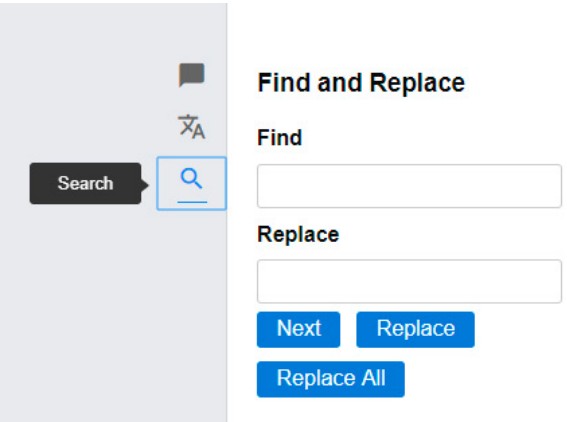

**Figure 2.** Interface of our initial prototype showing the Edit view with additional features.

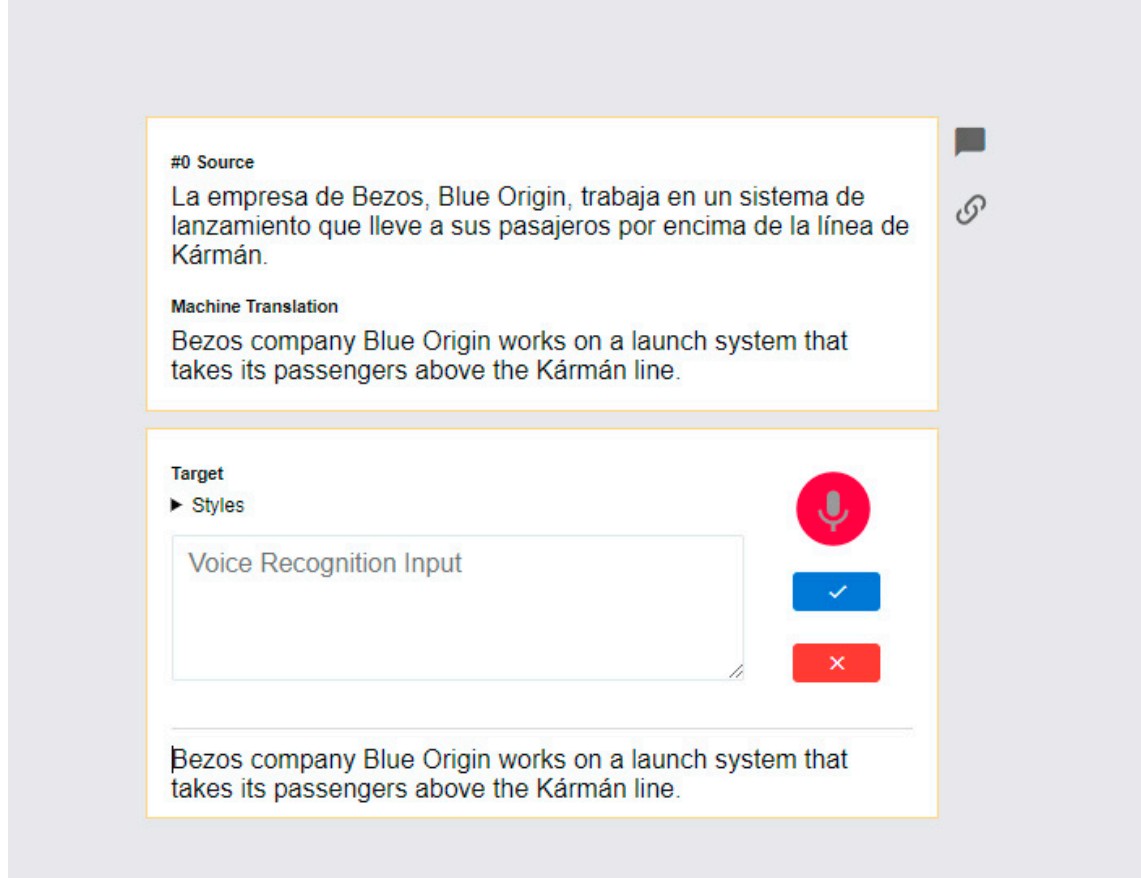

**Figure 3.** Interface of our initial prototype showing the Edit view with the dictation box active.

Finally, Figure 4 shows the Tile view, where users may select one or more words and move them around. They can also double-click any tile to edit or delete its content. To add a new word, users must double-click the tile before the desired insertion point, move to the final position, type a space, and then type the new word. Hitting Enter accepts the changes.

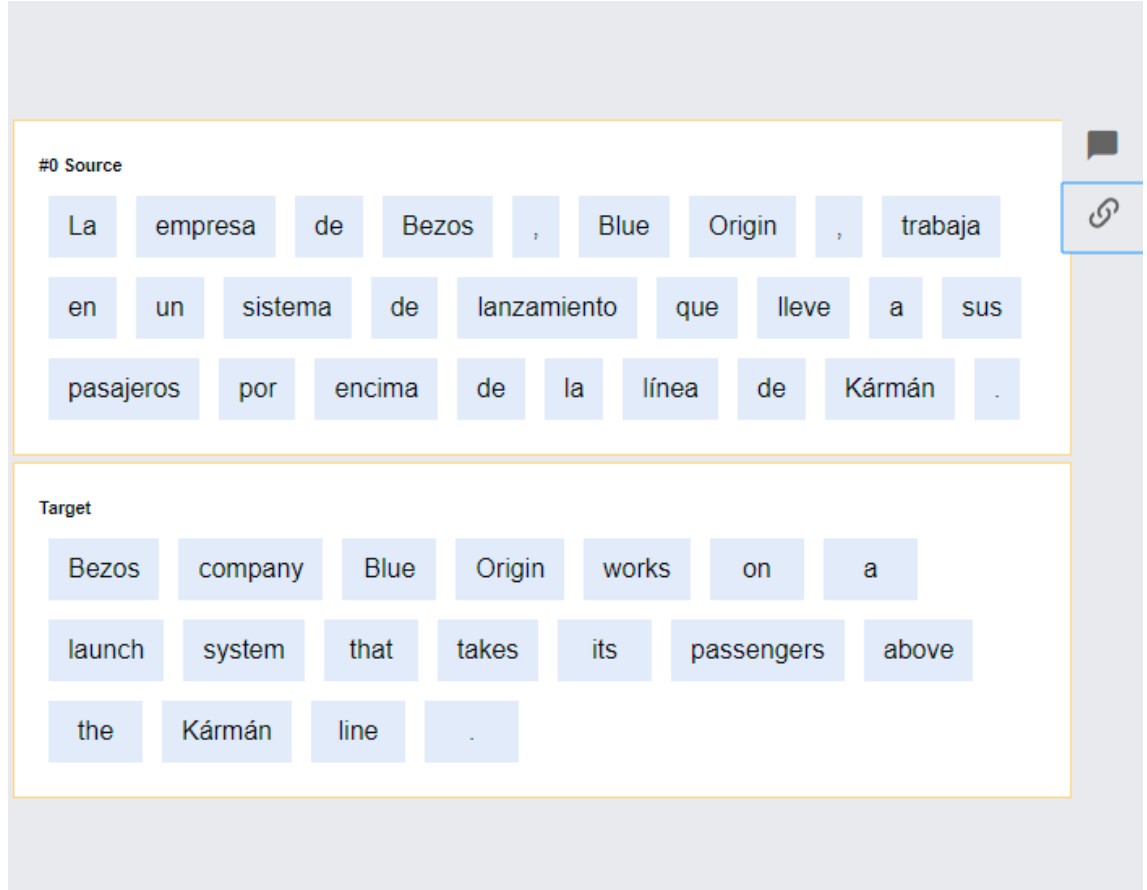

**Figure 4.** Interface of our initial prototype showing the Tile view, designed for touch interaction.

### 4.1. Testing the Initial Prototype

The initial prototype was tested for usability during October 2017. For this test we invited 10 professional translators based in Ireland as our participants. Six were French-to-English translators while four were Spanish-to-English translators. Three of them had tried translation dictation, but had not managed to use it consistently. The rest had no experience of ASR. All participants in this study gave their informed consent for inclusion prior to participation. The study was conducted in accordance with the Declaration of Helsinki, and the protocol was approved by the Ethics Committee of DCU (DCUREC2017_150). The demographic details of participants are provided in Figures 5–8.

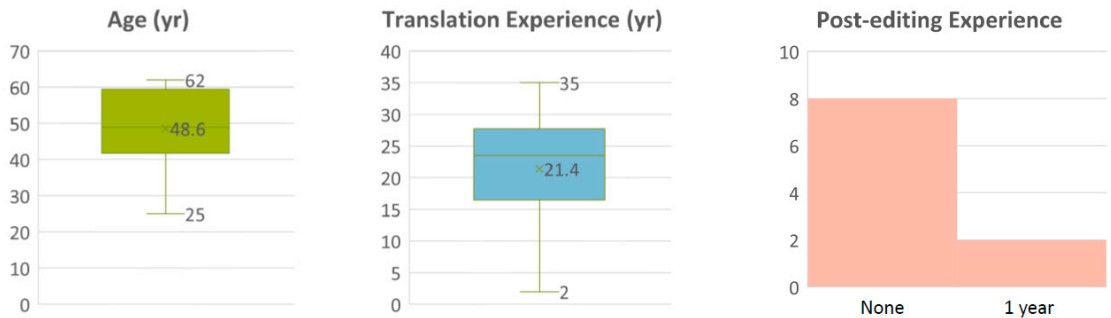

**Figure 5.** Participants' demographics: age, translation experience, and postediting experience.

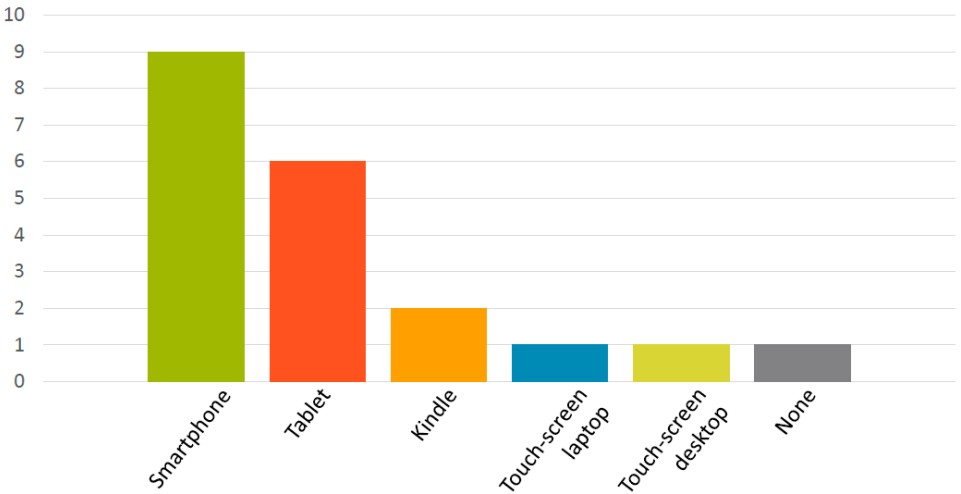

**Figure 6.** Participants' demographics: experience using touchscreen devices.

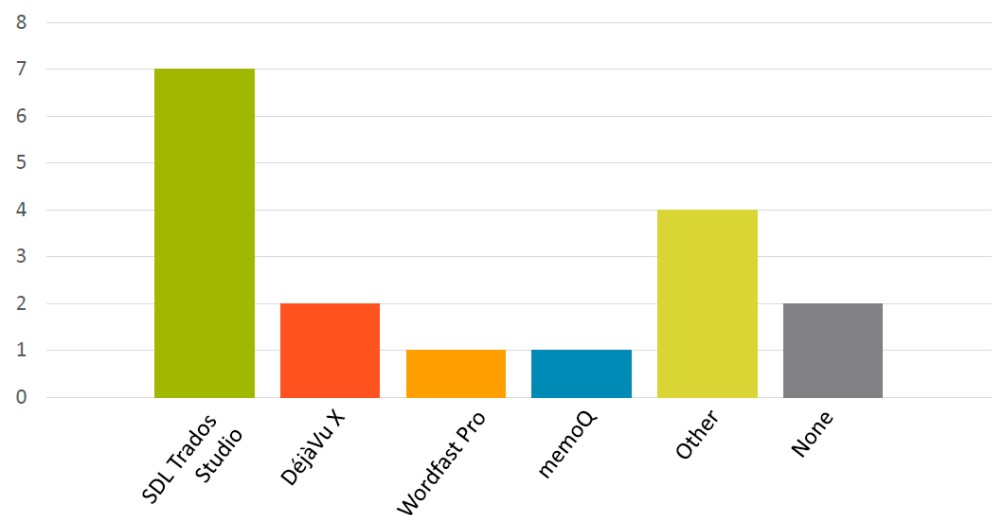

**Figure 7.** Participants' demographics: familiarity with Computer-Assisted Translation (CAT) tools.

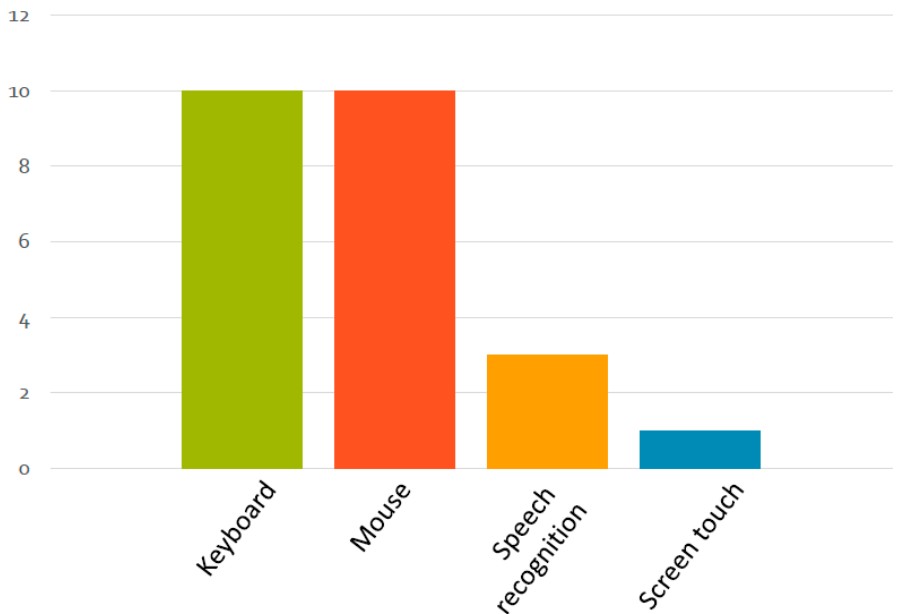

**Figure 8.** Participants' demographics: experience of input modes prior to the experiment.

Participants were asked to translate four texts, each containing between 303 and 321 words, extracted from a longer article in a multilingual corporate magazine. One set of texts was taken from the French edition of the magazine while the other set was taken from the corresponding Spanish edition. We chose to carry out the experiment in two language pairs that have tended to produce good quality MT output, in an effort to make our findings more generalizable. We decided to select the source texts in this manner to make sure they were perfectly comparable between the two languages, but it is worth noting that both source texts were actually translations of the same English original.

Both the French and the Spanish 'source' texts were pretranslated into English using the Google Translate neural MT system in October of 2017. Participants were then asked to perform four postediting tasks, each of them involving the following interaction modes.

- Keyboard & mouse
- Voice or touch
- Touch or voice
- Free choice of the above

In the first task, all participants used the typical keyboard & mouse combination; in the second and third tasks, they were asked to use the ASR feature for dictating their translations (or part of it) or to use the touch interaction (the order between the second and third tasks was alternated between participants); and in the fourth task they could choose to use any of the previous methods or a combination of them.

Data was collected using the tool's internal logging feature, screen recording software (Flashback Recorder) and an additional keystroke logging tool (Inputlog), for redundancy purposes. Upon completion of the four translation tasks, participants were asked some questions as part of semistructured recorded interviews.

*4.2. Results*

The numerical results obtained in the initial test are shown in Figures 9 and 10. Edit distance [22] is measured using the HTER (Human-targeted Translation Error Rate) metric to estimate the fewest possible steps from the initial MT suggestion to the final target text. Figure 9a indicates that translators tended to produce fewer changes when using voice dictation than when using the keyboard and mouse, and even fewer changes when using the touch mode, although our data is not sufficient for testing the statistical significance of those differences. Figure 9b indicates a similar pattern in terms of edit time, suggesting a correlation between the amount of changes and the time spent producing those changes, as expected. The error bars represent standard error values. Tile drags are only present in the touch interaction (tile view) and only occur when translators move words (tiles) around with their fingers. There were only 38 tile drags in total for the ten translators. This is a very low number of such movements, considering that translators handled 3100 words in total in that interaction (~310 words per translator). This already gives a hint about the restricted usefulness of the feature.

Figure 10 shows the number of characters produced (relative to the number of source words) in each interaction mode. The blue bars in the bar charts represent the sum total of the other bars.

The user feedback obtained from the post-task interviews is perhaps the most interesting and useful set of data for this type of usability experiment, as it gives clearer hints about certain phenomena and behaviors.

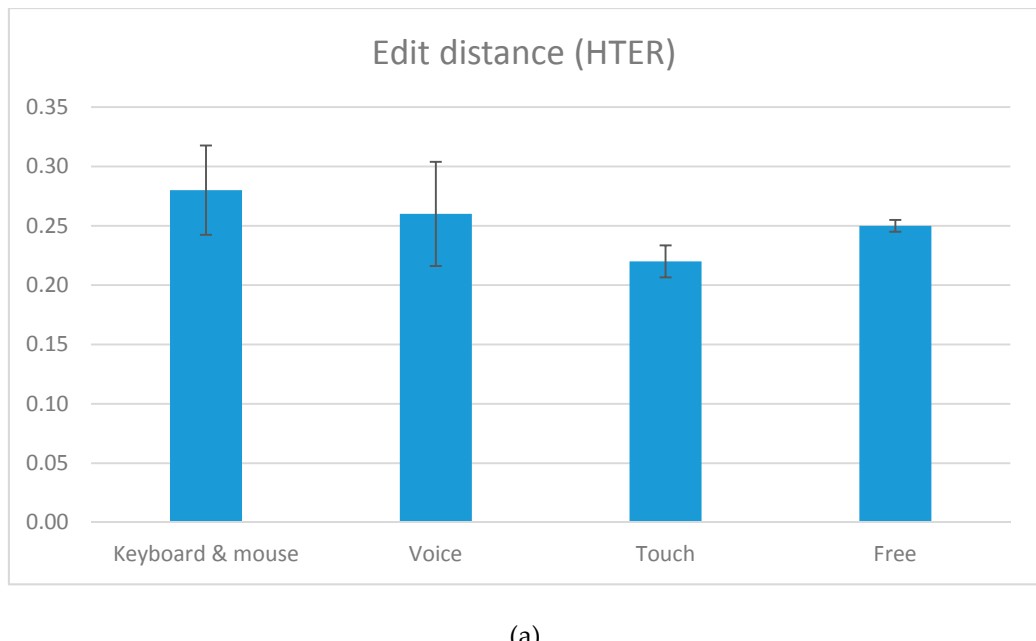

(a)

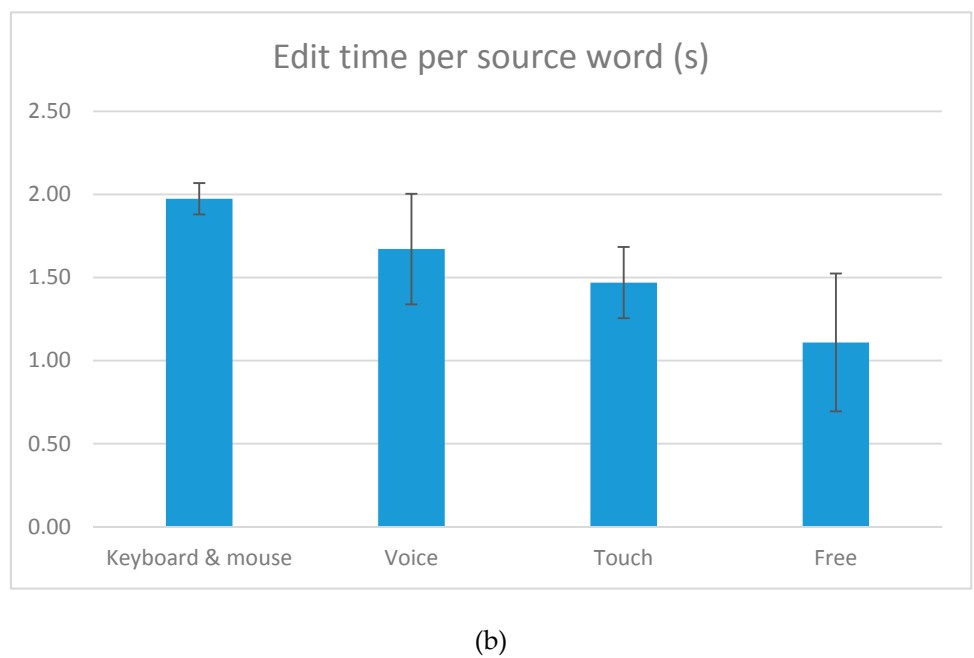

(b)

**Figure 9.** Results for edit distance (**a**) and edit time (**b**)—1st round of tests.

The first question asked to participants was "Which interaction mode did you prefer when performing the previous tasks?" As shown in Figure 11, eight translators preferred to work with the traditional input mode of keyboard and mouse, one preferred to work with ASR, and one preferred the combination of those input modes. Notably, not a single participant mentioned the touch (tile view) as one of their preferred interaction modes.

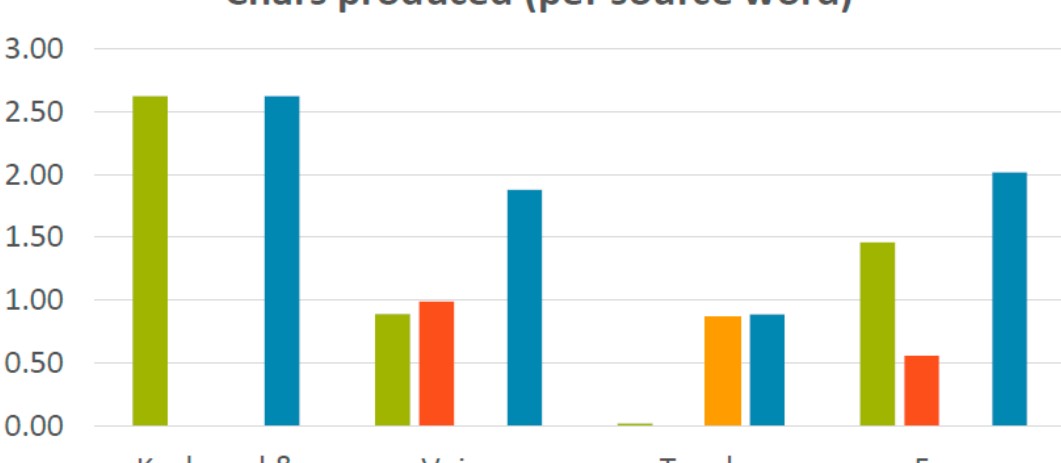

**Figure 10.** Number of characters produced in each of the interaction modes (means).

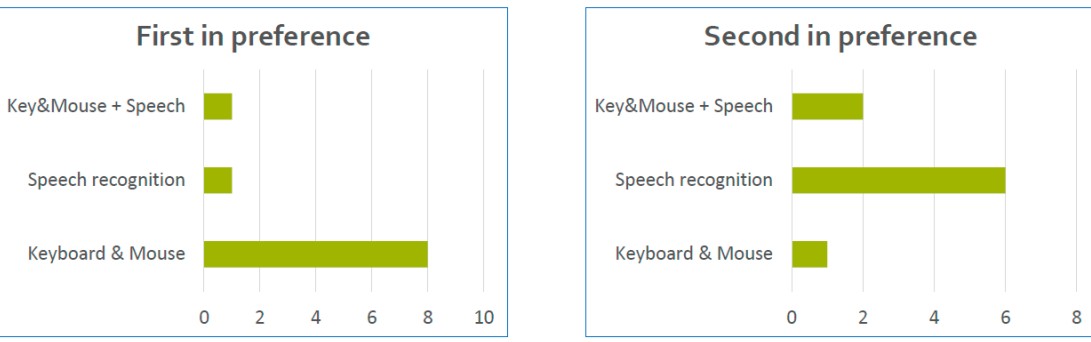

**Figure 11.** Preferred interaction modes as indicated by participants.

We then asked participants about the problems they encountered when using the different interaction modes. The list below includes all the points raised by our participants, with the numbers in brackets indicating how many participants mentioned a particular issue.

- General

  ○ Position of target text box (1)

- Speech recognition

  ○ "Did not pick up what I said (sometimes)" (6)
  ○ Placement of words & spacing (3)
  ○ Punctuation (2)
  ○ Capitalization (2)

- Touchscreen

  ○ Tile mode breaks up the sentence: "makes it hard to read" (both source and target), "no fluidity", and "can't see the 'shape' of the sentence" (8)
  ○ Difficult to move tiles around (dragging not working as expected) (3)
  ○ "'Alien' interface", "Feels artificial", and "not intuitive" (3)

&#9675;      Slows you down (3)
&#9675;      "Unwieldy" (1)
&#9675;      "Frustrating" (1)
&#9675;      Not ergonomic (position of hands) (1)

- Other

&#9675;      General lack of familiarity with tool and equipment (4)
&#9675;      Segmentation (1)

The semistructured interview used to gather user feedback was based on questions that focused on the problems encountered by the participants, so it comes as no surprise that the list above includes only the weaknesses, not the strengths of the tool.

We also asked how the tool compared with other CAT tools they might have worked with previously. On the positive side, two participants mentioned our "minimalist interface" as an advantage. On the negative side, the lack of translation memory functionality was mentioned by two participants, as was the low availability of keyboard shortcuts (one participant).

We then gathered their suggestions for improving the tool, which are compiled in the list below.

1. Tile mode: Option to see source not as tiles
2. Increased accuracy for voice recognition
3. Option to join/split segments
4. Additional step for revision
5. Auto-suggest feature
6. More shortcuts

The final two questions addressed the validity of our experimental design. The first was "Did you find the text difficult?" Nine participants answered negatively, saying that the text was "good" or "fairly straightforward". One participant said that "the beginning was more difficult than the rest", which might be the result of getting used to the research environment. The second question was "What did you think of the quality of the machine translation?" Nine participants said that this was "very good", "quite good", "pretty good", or "really good", while one answered "reasonably good".

In summary, we concluded that the editing environment had been well-accepted and the voice interaction was "surprisingly good" and useful for (some) participants. As for the touch interaction, the list of problems was quite long, but we believed it had potential for being further explored.

## 5. The Second Prototype

Based on the feedback received on the initial prototype, there were decisions to be made as to how to proceed. In general terms, we decided to focus on improving and retesting the existing features rather than implementing and testing new features. For the general Edit view, where translators interacted with the tool using the keyboard and mouse, the main change required was to give more prominence to the target text box and bring it closer to source text box. For the speech interaction, we focused on improving the automatic capitalization and spacing, as well as the handling of punctuation (brackets, apostrophes, etc.). Finally, for the touch interaction, we decided to completely redesign the Tile view and reconsider its use cases.

The only new feature that was implemented between the two iterations was the addition of voice commands (select, copy, cut and paste text, move to next segment, etc.). Eventually, however, we decided not to include them in the tests, because the response times obtained in our own pretests was too long to make the feature usable. It is worth considering a local installation of an ASR system (such as Dragon Naturally Speaking) rather than the Google Voice API to reduce the lag enough to make this function a useful addition.

Although there were also some visual changes to the Edit view and the dictation box, the Tile view underwent considerable redesign. Figure 12 shows how the interface looked after the changes were implemented. On the left side is a context bar, where translators can see the source and target texts for the current and surrounding segments. This was introduced to account for a common complaint from translators who said that it was difficult to read the texts as tiles. Improvements were made to the look and feel of individual tiles, and the need to double-click to edit was changed to single-click. While in the past translators were required to select the text in a tile and hit the Delete key and then Enter to delete a tile, now there was a 'bin' onto which they could simply drag the tile to be deleted. In addition to this, more buttons were introduced, namely for adding a tile, for undoing an edit, for redoing an edit, for accepting a segment, and for clearing a segment. On the left side of the active segment, new buttons were also introduced for quickly capitalizing or decapitalizing words. Finally, the visual response to the dragging of tiles was improved, to make it easier to identify where the moving tile was going to be inserted. It is also worth noting the substantial effort involved in improving the tokenization.

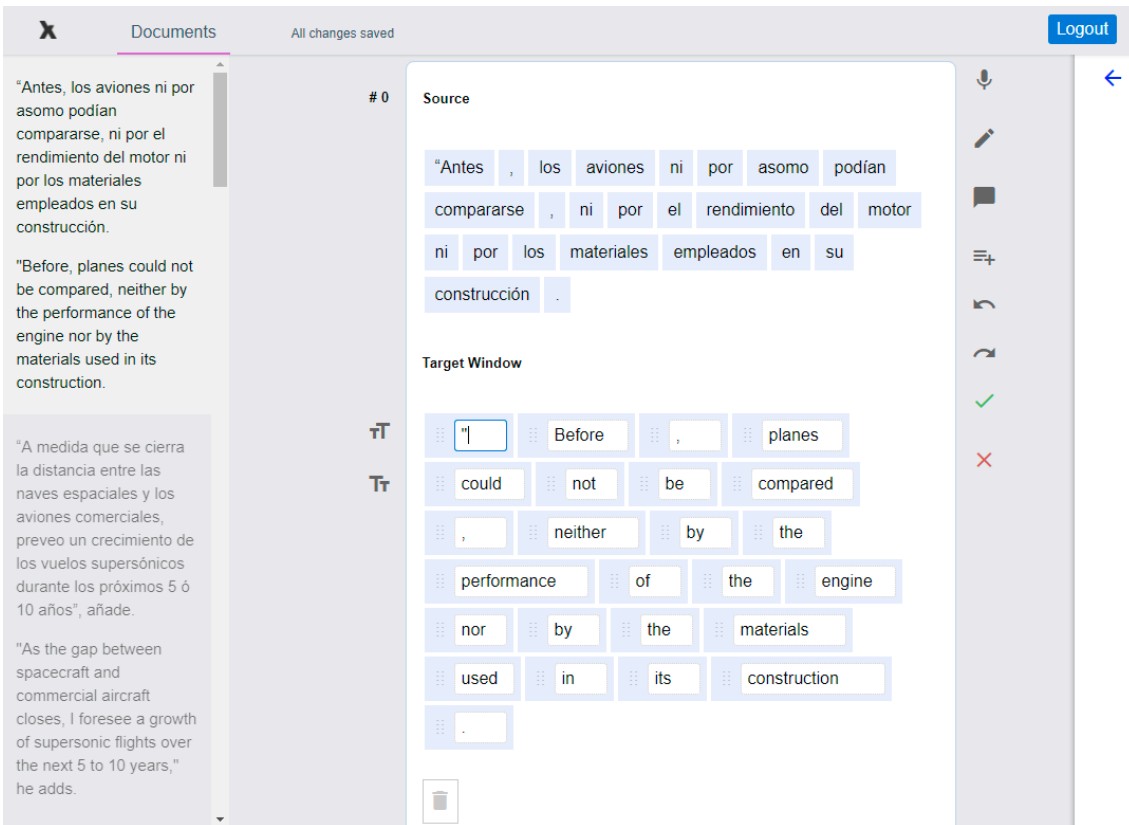

**Figure 12.** The Tile view in the second iteration of the tool.

Figure 13 shows the new look of the Edit view with the dictation box activated. One of the main changes is the prominence and position of the target text box, which now comes immediately below the source text box. This incorporates the feedback we received from translators in the initial tests, and makes the position of screen elements more consistent, considering that in the future we plan to offer more than one translation suggestion. These suggestions, either from translation memories or alternative MT engines, can now all be displayed below the target text box.

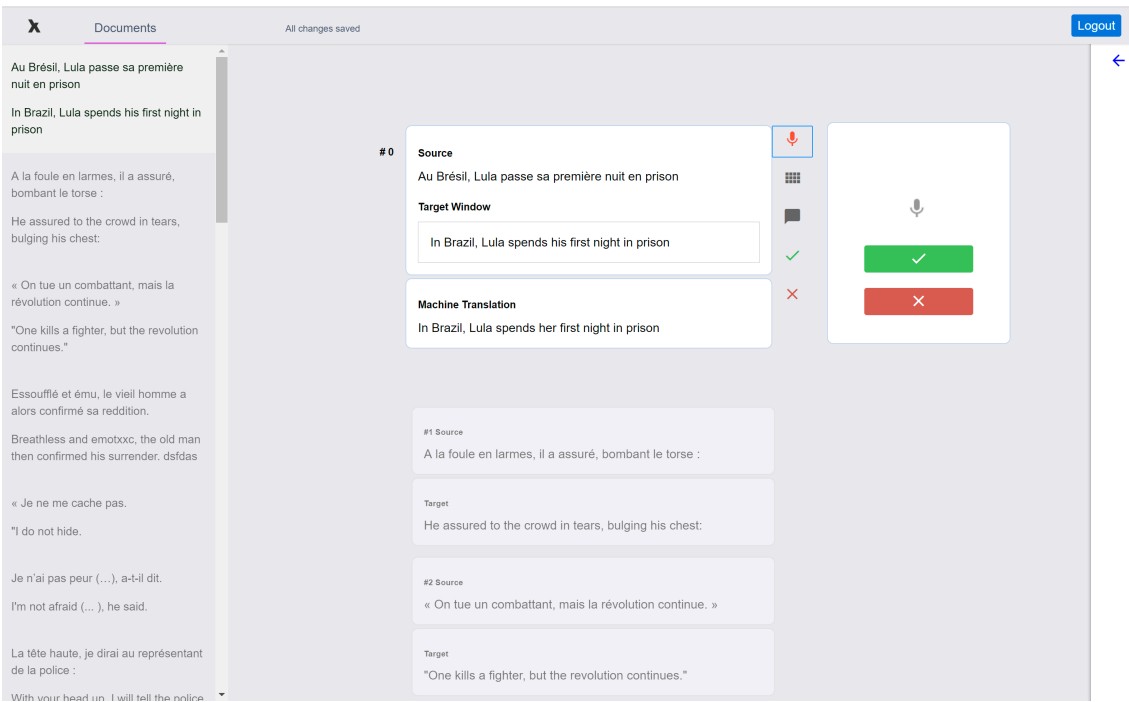

**Figure 13.** The Edit view in the second iteration of the tool.

Another change that may be noticed is the absence of the horizontal 'Styles' bar that used to be present at the top of the target text box to allow for text formatting. Although this feature is still available, we decided to hide it for the tests as this was not one of the features we wanted to test. New buttons were introduced on the right-hand side of the active segment, namely a green tick to accept the segment and a red X to erase the content of the target segment (although users could use standard CAT keyboard shortcuts). Another change was the position of the dictation box, which now appears to the right of the segment, instead of below it. Finally, the same context bar that is present in the Tile view is also present in the Edit view.

*5.1. Testing the Second Prototype*

The second prototype was tested for usability during April 2018. The methods for the second tests were quite similar to those used for the first tests, with some minor changes. Our participants were still Irish professional translators working from French and Spanish into English, but this time we had just eight participants instead of ten, five of whom had been participants in the first round. One of them had used translation dictation, while the rest had no experience of ASR.

We again used four texts of around 300 words each pretranslated with Google neural MT in April 2018, but instead of taking an article from a corporate magazine and splitting it into four chunks, this time we took four articles from newspapers in Spanish (El País) and French (Le Monde) as our source texts. The articles were from the same day and on similar topics: the recent imprisonment of Brazilian president Lula, the most popular series on Netflix and the streaming-giant's business model, environment-related topics, and issues related to Chinese mobile phone manufacturers in the USA. Although the source texts cannot be said to be perfectly comparable like those in the tests of the first prototype, this time they were original texts in the respective languages, rather than translations of the same original. This change in the approach for selecting the source texts was introduced to account for the feedback received from one participant, who said they had the feeling they were back-translating, i.e., the English source text had some feel of a Spanish original.

One major change introduced was the way that translators were asked to use the computer for the touch interaction. When testing the first prototype they used a Microsoft Surface Book in laptop

mode, while in the second test they used the same equipment but in tablet mode. This was done to prevent translators from instinctively returning to the keyboard and mouse and to create a scenario that was more similar to the typical scenario for the use of touchscreen devices, such as tablets and smartphones. The data collection methods were identical to the ones used in the first round, as were the number, type, and order of tasks the participants were required to perform.

*5.2. Results*

Figure 14 shows the numerical results of the tests with the second prototype. The results for edit distance indicate an even more pronounced difference between the touch interaction (tile view) and the other modes when compared to the first round of tests (cf. Figure 9). It shows that translators introduced fewer changes to the MT suggestions when using the second prototype than when using the first one. These are presented with the caveat that the number of participants were few, and that there was a good deal of individual difference, so while the average HTER per input type as may be seen in Figure 14 are 0.34, 0.36, 0.2, and 0.26, standard error (as shown in the error bars) was 0.06, 0.06, 0.03, and 0.07, respectively. Figure 15 shows the characters produced for each input type. Where there had been 38 tile operations in total in the previous round, there were 233 tile operations in this iteration, suggesting improved usability of touch interaction.

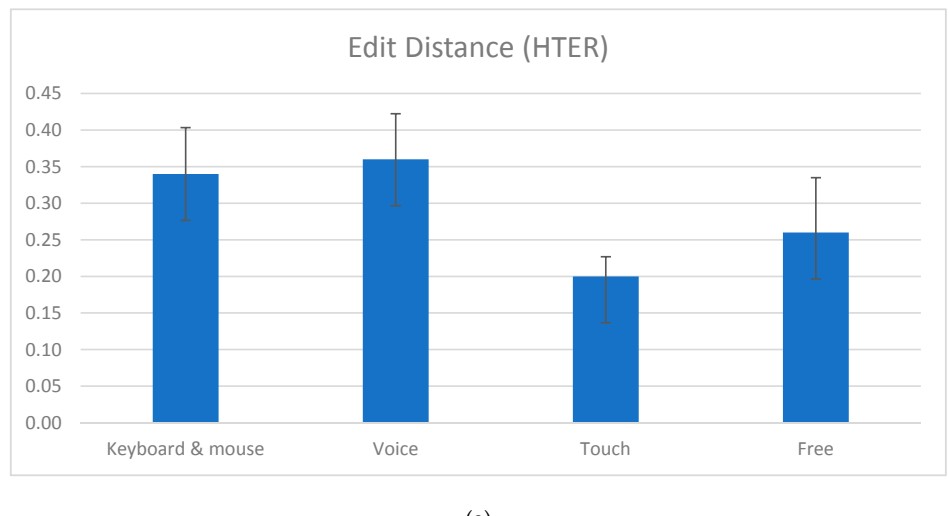

(a)

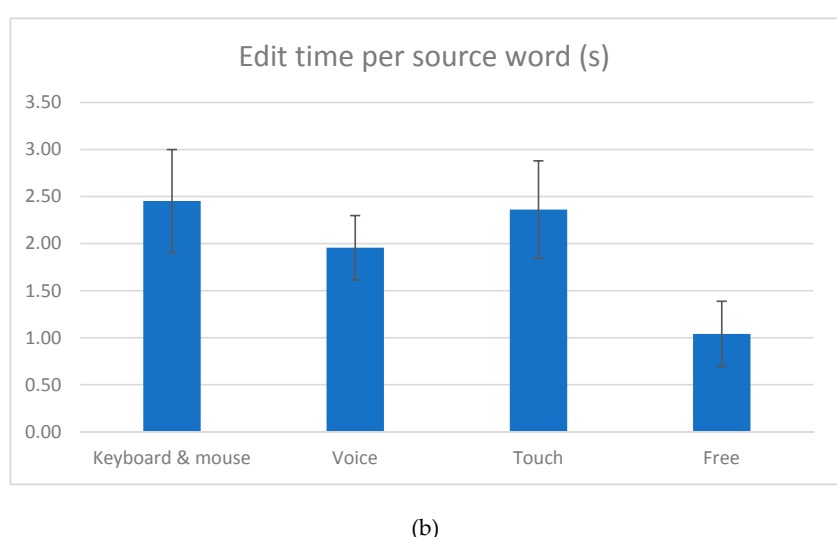

(b)

**Figure 14.** Results for edit distance (**a**) and edit time (**b**)—2nd round of tests.

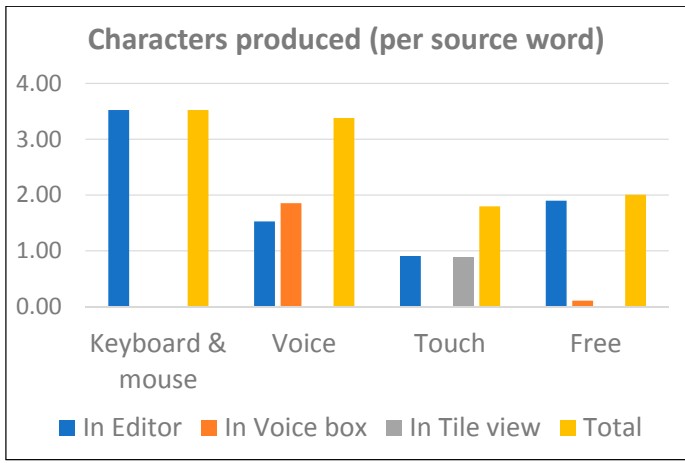

**Figure 15.** Number of characters produced in each of the interaction modes—2nd round of tests.

If we look at the results for edit time, translators spent the longest time when using the keyboard and mouse, closely followed by touch, while dictation was the fastest interaction. Again, our data are not sufficient for testing for statistical significance of these differences, but there seems to be an indication that the touch mode required more time to produce fewer changes, compared to the other interaction modes. As the source texts used in the touch interaction were of equivalent difficulty to the texts used in the keyboard and voice interactions and the MT engine was the same for all texts, the smaller number of edits needs to be explained by other factors, which we will consider below.

For the second round of tests, we included quality evaluation in our analysis. The evaluation was performed by one of the researchers, and no error taxonomy was adopted a priori; instead, error categories were created as errors were being found. It is worth noting that we only flagged the more "formal" errors, so there are no errors related to style or fluency in the list, as shown in Table 1.

**Table 1.** Types of errors encountered in the final translations, according to the type of interaction.

| Keyboard & Mouse Task | Number of Errors |
|---|---|
| Grammar | 2 |
| Omission | 2 |
| Spacing | 7 |
| Typo | 2 |
| Typo (leftover) | 1 |
| Total | 14 |
| **Voice Task** | |
| Grammar | 1 |
| Meaning | 2 |
| Omission | 2 |
| Proper noun | 1 |
| Punctuation | 3 |
| Spacing | 7 |
| Typo | 3 |
| Total | 19 |
| **Touch Task** | |
| Capitalization | 1 |
| Grammar | 6 |
| Meaning | 3 |
| Misplacement | 3 |
| Punctuation | 1 |
| Repetition | 1 |
| Spacing | 7 |
| Typo | 3 |
| Total | 25 |

The result of the quality evaluation is also provided in Figure 16. Those values are the aggregated total numbers of errors divided by the aggregated total number of words, normalized per thousand words; therefore, they do not account for individual differences between translators.

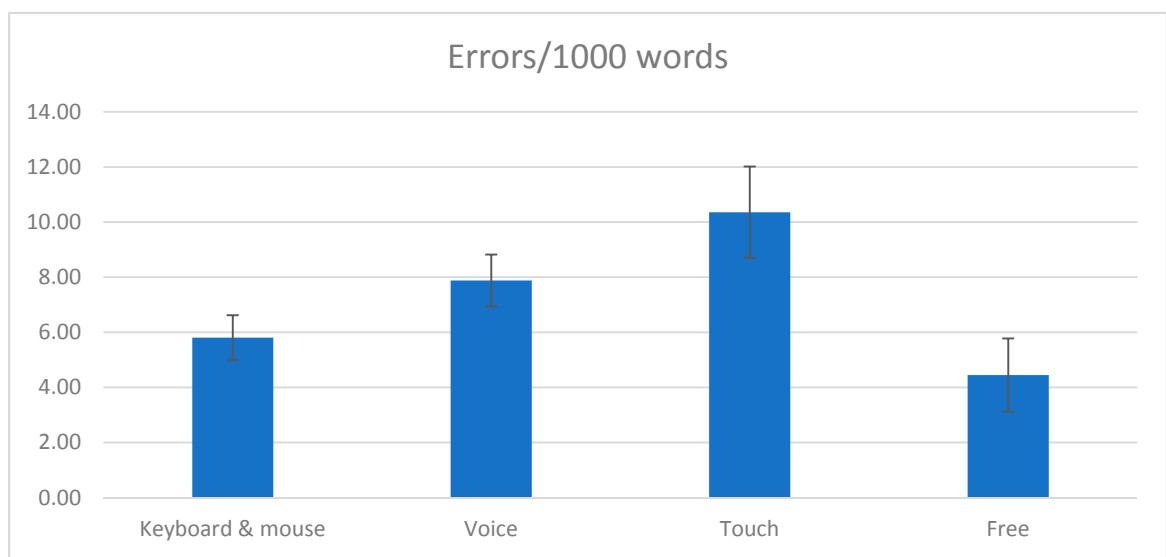

**Figure 16.** Results of the quality evaluation—2nd round of tests.

The results seem to indicate that the relatively small number of changes made in the touch mode resulted in more errors in the final translation. Therefore, a possible explanation for the smaller edit distance and number of characters produced in the touch mode as compared to the other interaction modes might be simply the difficulty of producing changes, rather than the absence of any need for those changes. However, before considering this further, let us look at the qualitative results gathered from the post-task interviews.

User Feedback

When asked about their preferred input modes, this time around four translators preferred the keyboard and mouse, while another four preferred the combination between keyboard and mouse + ASR. One translator liked the Tile view, and one liked the possibility of having a touchscreen, but not the Tile view.

Even though the interaction with keyboard & mouse (in Edit view) was not the focus of our development efforts between the first and second iterations of the prototype, the small glitches fixed and the few visual changes introduced were well received, including some positive comments on the usefulness of the new Accept button. Two participants complained about the lack of a spell-check function (or defective spell-checking), while one participant complained about the lack of context (in that they found it difficult to visualize the entire surrounding segments or to navigate through the file without losing track of the active segment).

As far as the voice interaction is concerned, in addition to the visual improvements, we fixed some glitches with automatic spacing and capitalization. Four participants spontaneously mentioned that the interaction was "intuitive" and "nice", and that it was "good to see the text before inserting" (as compared to other tools that insert the dictated text directly at cursor position in the target text box). The complaints we received were the following (each of them mentioned only once):

- Sometimes words are inserted in the wrong place;
- The default voice dictation box size is too small;
- The button to accept the result of voice recognition in the voice dictation box is too similar to the button to accept the entire segment, which can cause confusion;

- "You have to know what you're going to say, have a pad on the side" (training with sight translation);
- It feels awkward to say just one word, it is easier to write it;
- It slows you down.

We also received some complaints about the automatic speech recognition (ASR) (the numbers in brackets indicate the number of participants who mentioned a specific issue):

- "Did not pick up what I said (sometimes)", "did not recognize my accent" (5)
- "You need to adapt your way of speaking" (1)
- "Especially bad for single words" (1)
- Does not recognize proper nouns (Antena 3, Huawei) (2)
- "Tricky with punctuation", capitalization, etc. (2)

Although we understand the frustration expressed by the translators, our tool is just the gateway to the ASR system, which we access using the vendor's API (in this case, Google Voice). Accordingly, we have no capability of fixing the issues that are associated with the ASR engine, and our focus has been on the behavior of the interface. Although Google Voice is one of the best general-purpose ASR systems available, in future tests we plan to connect to customizable ASR systems (such as Nuance's Dragon Naturally Speaking), which can be trained separately for each participant. We also acknowledge that using an ASR system effectively requires some previous training. Recognition of proper nouns is a common complaint when using ASR, and was actually one of the motivations for incorporating touchscreen functionality, as this allows users to drag proper nouns directly from source to target. Finally, it is worth mentioning that the other three translators felt that the ASR system recognized what they said "well" or "very well", and that one translator felt that voice dictation "allows you to type less".

Looking at the feedback on the touch interaction, this was again the area where more issues were raised. The positive comments were the following.

- "On-screen keyboard was good" (4)
- "Tablet mode OK" (2)
- "Responded well" (1)
- "Handier for changing things [as opposed to using the keyboard and mouse]" (1)
- "Automatic capitalization good, choice to change it manually using the new buttons is also good" (1)
- Visual improvement from last time ("looks nicer") (1)

On the other hand, the list of problems mentioned by the participants is still very long:

- Tiles take too much space on the screen, "Spread-out text, Hard to read the sentences (ST and MT, and the changes I've made)" (6)
- "Bin not working as expected": slow, did not delete, wrong words being deleted (4)
- Context band too narrow (1)
- Unable to select multiple tiles (words), interrupts the flow of work
- "Wouldn't obey", "have to type hard", "difficult to place word where you want"

It is interesting to note that the context band, which was introduced in response to feedback received in the first test, where participants mentioned the difficulty of seeing the source and target texts in context, did not have the effect we expected. This might have been due to users' lack of familiarity with the tool, as only one participant confirmed that they used it, three participants said they had not used it, and the remaining four participants were not sure. Of course, it might also have been due to ineffective design on our part.

Finally, we also heard some general negative comments about the Tile view, such as

- "Hated it", "Didn't like it at all", "It was awful" (4)
- "Frustrating", "not enjoyable" (2)
- "Difficult to use" (1)

## 6. Discussion

The popularity of automatic speech recognition (ASR) for translation is growing [23], and this mode of interaction seems to have worked very well for some translators, but not so well for others. Most of the issues raised are related to the ability of the ASR engine to properly recognize speech, rather than to the usability of the tool's interface, which has been our main focus of research and development. We believe that this can be remedied by training translators to work with ASR systems as well as by using customizable ASR systems that can adapt to individual voices. Another area that remains to be explored is how to best utilize ASR in combination with translation memories (TM) and machine translation (MT) [8], since the suggestions offered by those technologies often require minimum intervention, and ASR tends to work better for longer dictations.

The touch interaction mode did not produce the results we expected. This appears to be not so much due to the usefulness of the touchscreen for interacting with the tool, but because of the Tile view that we designed for this interaction. Initially we wanted to use the Editor view for seamlessly combining all interaction modes, but due to the limitations associated with HTML5 (browser-based applications) and how operating systems capture signals coming from the touchscreen (which is different from a mouse) we needed to find an alternative. Those limitations include the inability to drag words with the finger as you do with the mouse, the behavior of pop-up menus, etc.

We had decided to focus on touch since the beginning of this project and then decided to improve it because of the potential offered by touchscreens on mobile devices and because it remains underexplored for professional tools and on desktop devices. We were ultimately attempting to create a Natural User Interface (NUI) as described by Wigdor & Wixon [24], where touch could be used in conjunction with other input modes. However, as those authors acknowledge, touchscreens might not be the best way of producing text [9]. The need to move text from one place to the other, which was the rationale for the tile drag feature, seems to have been more prominent in previous MT paradigms (rule-based or statistical MT) than it is in the current neural MT paradigm, considering the small amount of such operations carried out by our participants. As Alabau & Casacuberta [18] conclude for their work with the e-pen, our touch mode might be better suited for situations where only a few changes are required, such as "postediting sentences with few errors, [repairing] sentences with high fuzzy matches, or the revision of human postedited sentences." It remains to be explored whether it can still be useful for certain use cases, such as for dragging proper nouns that are out of the vocabulary of the ASR system from source to target.

### 6.1. Limitations

In a later stage of our study, we plan to analyze the errors in the MT output so that the remaining errors in the final translations can be put in perspective. A similar analysis should be done with the recognition errors made by the ASR system. Those two factors can certainly affect the number of edits and the time needed to correct the suggested translation, as well as the perception of the system that we collected from our participants.

### 6.2. Next Steps

The tool includes several features that have not been tested yet, as the focus during this first phase of development has been on the different interaction modes. Those additional features include the ability to

- handle formatting using xml tags;
- join and split segments;

- add comments;
- search & replace text;
- look up terms in a lexicon; and
- use voice commands.

In addition to these already-implemented features, we aim to include provision for handling TMs, with fuzzy matching and concordancing. Another improvement that is being considered for future releases is the ability to handle more file formats, in addition to the existing XLIFF-only support.

Considering the substantial efforts involved in resolving the problems associated with the touch interaction, our next priority will be to test the system with a locally-installed ASR system, which will hopefully make the response time short enough to turn voice commands into a viable alternative.

Finally, we aim to improve the usability of the ancillary screens, namely the log-in and sign-up pages as well as the file management page.

*6.3. Web Accessibility Tests*

In a subsequent stage, we conducted usability studies with three professional blind translators to assess the effectiveness of the web accessibility features that have been incorporated into the tool. The results will be reported in a separate publication and were very encouraging, pointing to one of the most distinctive features and promising applications of our tool.

## 7. Conclusions

This article reports on the development and testing of a translation editing tool, building on our team's experience of assessing interface needs [1], testing usability of desktop editor features [4], and developing a voice- and touch-enabled smartphone interface for postediting [2]. We carried out iterative usability tests with professional translators using two versions of the tool for postediting. The results include productivity measurements and satisfaction reports and suggest that the availability of voice input may increase user satisfaction. Participants liked the minimal interface and found interaction with the tool to be intuitive. While touch interaction was not initially well-received by translators, after some interface improvements it was considered to be promising, although not useful in its current implementation. We believe that touch may need to be better integrated to provide a valuable user experience. The findings at each stage provided feedback for subsequent stages of agile, iterative development.

Several authors have suggested that error rates for ASR can be reduced by using data from the source text through MT models. This is an avenue worth exploring in future studies, although it will require a different approach than the one we used, namely to connect to a publicly available ASR API. As mentioned previously, our goal was to explore the user experience of using the tool interface, rather than to explore ways of improving ASR or MT performance, although we acknowledge that the former is dependent on the latter.

Ultimately, we will test whether nondesktop large-scale or crowd-sourced postediting is feasible and productive and can help alleviate some of the pain points of the edit-intensive, mechanical task of desktop postediting.

**Author Contributions:** Conceptualization, J.M. and C.S.C.T.; Methodology, C.S.C.T.; Software, D.T. and J.V.; Validation, C.S.C.T., J.M., and J.V.; Formal Analysis, C.S.C.T. and J.M.; Investigation, C.S.C.T.; Resources, A.W.; Data Curation, C.S.C.T.; Writing—Original Draft Preparation, C.S.C.T.; Writing—Review and Editing, J.M., C.S.C.T., and A. W.; Revision C.S.C.T and J.M.; Supervision, J.M. and J.V.; Project Administration, J.M.; Funding Acquisition, J.M. and A.W.

**Funding:** This research was funded by Science Foundation Ireland and Enterprise Ireland under the TIDA Programme, grant number 16/TIDA/4232 and the ADAPT Centre for Digital Content Technology, funded under the SFI Research Centres Programme (Grant 13/RC/2106) and cofunded under the European Regional Development Fund.

**Acknowledgments:** We thank Sharon O'Brien, who initiated and collaborated in the design of our previous mobile postediting interface, the anonymous participants in this research, and the reviewers and editors of this special issue for their constructive comments and suggestions.

**Conflicts of Interest:** The authors declare no conflicts of interest. The funders had no role in the design of the study; in the collection, analyses, or interpretation of data; in the writing of the manuscript, or in the decision to publish the results.

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
