# Peer review of "Creating a Multimodal Translation Tool and Testing Machine Translation Integration Using Touch and Voice"

_informatics, doi:10.3390/informatics6010013_

Reviewer 1 Report

This a paper presents the prototype of a web-based translation editing tool that permits multi-modal input via touch-enabled screens and speech recognition in addition to keyboard and mouse. This approach to multimodality in translation software is original and the paper provides an advance in the translation technologies field. The prototype is well described, and testing is well designed and executed. Method, data and results are presented clearly and honestly (not hiding criticism from testers). Despite the relatively low number of participants, results are significant and useful to draw conclusions, improve the prototype and prepare for future testing. The methods, tools and testing are described with sufficient detail to allow further reproduction of results using the same software prototype. Figures are relevant and well-presented.

The article is written in an appropriate way and presentation is clear. 

The design of this prototype will attract the attention of readers not only interested in translation technologies in general but also readers interested in ASR (Automatic Speech Recognition). With this prototype the authors address an interesting feature related to machine translation post-editing: multimodal input, which is an underresearched area in translation technologies. 

Detailed comments: 

Page 1. Line 28. Introduction could be improved by including some more relevant references on the relationship between CAT tools and automatic speech recognition. 

Page 1. Line 32 – “for repairing translation memory (TM) matches” 

Reviewer’s suggestion: 

“for repairing translation memory (TM) mismatches” (??)

Page 2. Line 5. 2. “Materials and Methods”

Reviewer suggests to change the name of the section as it is mainly devoted to the description of the development of the tool and “materials and methods” are not clearly stated. 

Page 3. Line 7. 

Question: Where does the lexicon come from?

Page 4. Line 5. “Once the dictation has been completed, the result of the recognition appears in the corresponding area. If translators are satisfied with the ASR output, they can accept it by clicking the green tick; if they are not satisfied, they can click the X button to clear the ASR output and dictate again.” 

Question: Does it mean that the box with the dictation cannot be edited? Is it possible to edit the box or not?

Page 8. Line 4. “Figure 9 (C) indicates a surprising low number of such movements (38).”

Question: Is this number (38) the total for the 10 translators or the mean of the 10 translators?

Page 8. Line 8. “Figure 1. Results for edit distance, edit time and tile drags - 1st round of tests.” 

Renumber to “Figure 9”

Page 8. Line 9. “Figure 10 shows the number of characters produce...” 

Question: Is this the mean of the 10 translators?

Page 10. Line 6. “4. The second iteration”. 

Suggestion: Taking into account the name for section “3. The initial prototype” (page 2. Line 31), would it make sense to name section 4 “The second prototype”?

Page 14. Line 4. 

Please specify if number of errors are the total for the 8 translators or a mean

Any comments on individual differences between translators?

Page 16. Line 40. ASR in combination with TM and MT

Suggestion: “ASR in combination with Translation Memories (TM) and Machine Translation (MT)”...

Page 16. Line 33

Section 5.1. is missing

Page 17. Line 20. “more file formats, in addition to the existing 20 XLIFF support.”

Question: Is XLIPFF the only supported format? No comments on format are included in the paper. Which format was used for the texts of the tests?

Page 17. Line 32. Conclusion.

Present content of “Conclusion” could be easily added in the “Next steps” section. Please rewrite Conclusions by restating the aim of the paper and summarizing the main points of the testing for the readers in order to understand the contribution of this prototype to research. 

Author Response

Dear Reviewer,

Our response is detailed in the attached PDF.

Best wishes,

Joss and authors

Reviewer 2 Report

This is an interesting tool which may prove very useful for a variety of reasons. The results are promising, but limited. The authors point out themselves that more participants would have been necessary. It is a pity that this was not considered. It might also not have been ideal to use translations as source texts and two different target language given that the MT quality may be different depending on the TL. Regarding the results it would be a minimum to show standard errors - especially because there were considerable differences regarding satisfaction with the different modes. Also, it might have been a good idea to choose participants who have no experience with ASR. There are clearly quite a number of translators who use dragon and they may have benefited much more from the interface than those who have never or only for other purposes used ASR. One particular result seems to be buried in how the data is presented. It seems that the most efficient mode may have been the free mode. But it is not clear what participants did for what proportion of time. It would be very interesting to see a breakdown of what they did in this mode for Figure 1 - in the free mode, participants are nearly twice as fast as compared to the tradition keyboard only mode. Table 1 seems to be lacking the figures for the number of errors in the free mode. This is particularly surprising, because it is the mode in which the fewest errors were left in the text.

Author Response

Dear Reviewer,

Our response is detailed in the attached PDF.

Best wishes,

Joss and authors

Round  2

Reviewer 1 Report

I have thoroughly read the revised version of the paper and I confirm it has been improved with suggested changes. The author response document was clear and well justified. In the new version I have also read changes proposed (I guess) by reviewer #2 and changes appropriate and useful. From my point of view,  the final paper is ready to be published. 

Author Response

Dear Reviewer,

Many thanks for your comments.

Thanks,

Joss & authors

Reviewer 2 Report

This version is much better - there is only one tiny aspect which could be improved: in my previous review I suggested to show standard errors in the graphs. You do so in Figure 5, but not the others. You now detail the standard deviation in the body of the text for some results, which is helpful, but to see them in the figures would be extremely helpful, too. I suspect most figures are produced using Excel, so it should be easy to include either standard deviations or standard errors. 

Author Response

Dear Reviewer,

Many thanks for your comments. Error bars illustrating standard error have now been added to graphs.

Thanks,

Joss & authors